# Evaluation of Ex Vivo Shear Wave Elastography of Axillary Sentinel Lymph Nodes in Patients with Early Breast Cancer

**DOI:** 10.3390/cancers16244270

**Published:** 2024-12-22

**Authors:** Riku Togawa, Helena Dahm, Manuel Feisst, Peter Sinn, André Hennigs, Juliane Nees, André Pfob, Benedikt Schäfgen, Anne Stieber, Oliver Zivanovic, Jörg Heil, Michael Golatta, Fabian Riedel

**Affiliations:** 1Department of Obstetrics and Gynecology, Heidelberg University Hospital, Im Neuenheimer Feld 440, 69120 Heidelberg, Germany; riku.togawa@med.uni-heidelberg.de (R.T.); helena.dahm@med.uni-heidelberg.de (H.D.); andre.hennigs@med.uni-heidelberg.de (A.H.); juliane.nees@med.uni-heidelberg.de (J.N.); andre.pfob@med.uni-heidelberg.de (A.P.); benedikt.schaefgen@med.uni-heidelberg.de (B.S.); oliver.zivanovic@med.uni-heidelberg.de (O.Z.); joerg.heil@kse-hd.de (J.H.); michael.golatta@kse-hd.de (M.G.); 2Institute of Medical Biometry (IMBI), Heidelberg University, Im Neuenheimer Feld 130.3, 69120 Heidelberg, Germany; feisst@imbi.uni-heidelberg.de; 3Institute of Pathology, Heidelberg University, Im Neuenheimer Feld 224, 69120 Heidelberg, Germany; peter.sinn@med.uni-heidelberg.de; 4Department of Diagnostic and Interventional Radiology, Heidelberg University Hospital, Im Neuenheimer Feld 440, 69120 Heidelberg, Germany; anne.stieber@med.uni-heidelberg.de; 5Breast Unit, Sankt Elisabeth Hospital, Max-Reger Strasse 5-7, 69121 Heidelberg, Germany

**Keywords:** breast cancer, sentinel lymph node, ultrasound, shear wave elastography, lymph node metastasis

## Abstract

The pretherapeutic assessment of axillary lymph node status is crucial in staging early breast cancer and determining treatment. Clinically unsuspicious axillary lymph nodes are usually assessed via sentinel lymph node biopsy, with metastases detected in up to 20% of cases. Shear wave elastography, a non-invasive ultrasound technique, has shown potential in identifying metastatic lymph nodes based on their stiffness. This study evaluated the ability of shear wave elastography to differentiate tumor-free from metastatic sentinel lymph nodes ex vivo by comparing elastography measurements with histopathological results. A total of 168 sentinel lymph nodes from 105 patients were analyzed, detecting metastases in 17 cases (16.19%). Contrary to previous studies, this study did not find significant differences in the shear wave velocity measured in tumor-free and metastatic sentinel lymph nodes. Further research is needed to clarify the potential role of shear wave elastography in axillary staging.

## 1. Introduction

The axillary lymph node (LN) status in breast cancer patients is one of the most significant prognostic factors in breast cancer patients, strongly influencing both disease recurrence and overall survival. It plays a pivotal role in determining the course of treatment, including systemic therapies such as chemotherapy and hormone therapy, and the extent of surgical intervention [1,2]. The accurate evaluation of axillary LN involvement is essential for tailoring treatments to individual patients and minimizing surgical overtreatment. Axillary B-mode sonography needs to identify breast cancer patients who have an inconspicuous axillary LN status and thus do not benefit from extensive axillary surgery. Axillary lymph node dissection (ALND) is typically recommended for patients with axillary LN involvement, but it carries a higher risk of complications, including lymphedema, seroma, and long-term impairment of arm function. For patients with negative or inconspicuous nodal status, sentinel lymph node biopsy (SLNB) is the preferred procedure, as it involves the removal of fewer nodes and is associated with significantly lower morbidity [3,4]. SLNB has been widely adopted in clinical guidelines for its benefits, including reduced postoperative complications and quicker recovery times [5].

Since its recommendation by national and international guidelines, axillary B-mode sonography has been regularly used in clinical routines [5]. Still, its diagnostic accuracy is low, as the criteria for LN positivity have been proposed but not yet standardized [6,7].

The sonomorphological criteria for metastatic axillary LNs are eccentric cortical thickening, loss of fatty hilum, rounder shape, and irregular margins [8,9,10,11]. These criteria have a low positive predictive value since these sonomorphological signs are unspecific and can also be seen in inflammation or reactively changed LNs. This has led to wide variability in the reported sensitivity and specificity of conventional sonographic B-mode axillary staging, ranging between 45 and 95%, respectively [12,13]. Because of this variability, the diagnostic workup of suspicious LNs identified on sonography typically requires confirmation through fine-needle aspiration or core needle biopsy, both of which provide more reliable results with specificity rates as high as 100%, although sensitivity still varies between 25% and 94% [14,15]. Despite the availability of biopsy confirmation, up to 20% of breast cancer patients with clinically negative LNs are later found to have metastatic involvement in the final histopathological analysis of the SLNB [12,16,17].

This highlights the need for additional diagnostic tools to improve the diagnostic performance of axillary ultrasound. In recent years, the use of shear wave elastography (SWE) has been studied as an additional ultrasound tool for the non-invasive assessment of tumors in the breast parenchyma and axillary LNs. SWE measures the stiffness of a lesion based on the shear wave velocity, which can be mapped to create a color-coded image of the stiffness in the region of interest and allow a quantitative measurement [18,19,20]. Increased tissue stiffness is a known predictor of malignancy. High stiffness is reported to be due to high levels of collagen, myofibroblasts, angiogenesis, inflammatory reaction, necrosis, and different tumor histologic biomarkers [21,22]. SWE has already been successfully integrated into breast diagnostics, and its use has been endorsed in the latest version of the BI-RADS atlas for assessing breast lesions. However, its application to axillary LNs remains investigational, and it has not yet been adopted as a clinical standard [23]. Early studies have shown that SWE is also feasible in axillary LNs and can provide valuable quantitative data that complement conventional B-mode sonography [24]. Currently, the data on the performance of SWE in the assessment of axillary LNs is still scarce. However, it has already been shown that SWE can be reliably performed in axillary LNs regardless of the examiner’s experience [24]. There is a lack of comprehensive systematic research on the use of SWE in the axilla, although there is a great need to evaluate non-invasive methods to optimize the assessment of axillary LN status. It has already been shown that SWE has the potential to differentiate between benign and malignant histology in sonographically suspicious axillary LNs with a sensitivity of up to 69.8% and a specificity of 87.5% [25,26,27,28]. Nonetheless, performing SWE in the axilla presents certain challenges. The anatomical complexity of the axillary region, including structures like the humeral head, clavicle, axillary artery, and vein, can make it difficult to obtain standardized and reproducible images. Additionally, variations in patient anatomy and excessive or inconsistent pressure applied by the ultrasound probe could artificially alter tissue stiffness and thereby limit the reliability of the results. As a result, there is ongoing research into improving the reproducibility and accuracy of SWE in this setting [29]. One innovative approach to overcoming these challenges is the use of ex vivo SWE, which involves performing the elastography assessment on surgically excised LNs immediately after removal [30,31]. This method allows for greater control over variables such as probe pressure and anatomical interference by providing more standardized conditions, potentially reducing the likelihood of artifacts and improving the accuracy of stiffness measurements.

In this single-center prospective explorative study, the feasibility of SWE on ex vivo axillary LNs will be evaluated in a clinically routine setting. It should be extrapolated if ex vivo SWE has the potential to differentiate between tumor-free SLN and LN metastasis.

## 2. Materials and Methods

### 2.1. Study Design and Enrolment

This is a single-center prospective diagnostic study. The study protocol was approved by the local ethics committee and additional written informed consent was obtained from each participating patient (S-320/2023). The study was conducted in a specialized and certified diagnostic breast unit and included *n* = 110 consecutive patients who underwent guideline-concordant SLNB as part of their primary surgical therapy of histologically confirmed breast cancer (including carcinoma in situ) between July 2023 and March 2024. All patients underwent previous axillary ultrasound staging with B-mode sonography and only those with no sonographic signs of pathological LNs (classified as cN0) were included. The exclusion criteria were male sex, patients younger than 18 years of age, previous ipsilateral axillary surgery (i.e., history of SLNB or ALND), prior radiation therapy to the ipsilateral breast, and patients who are pregnant or lactating. Following surgical excision, all fresh SLN specimens were evaluated with SWE ex vivo in an unfixed condition right after removal. Subsequently, the specimens were sent to the pathology department for routine histopathological examination. To ensure accuracy, each SLN was individually processed, allowing for a precise one-to-one correlation between the SWE findings and the histopathological results.

### 2.2. Shear Wave Elastography (SWE)

SWE was performed using 2D-SWE systems Siemens Acuson S2000 or S3000 equipped with virtual touch tissue imaging quantification (VTIQ) software (https://www.siemens-healthineers.com/en-us/ultrasound/tissue-strain-analytics/virtual-touch-iq, accessed on 13 December 2024), utilizing a 9MHz probe (Siemens Healthineers, Erlangen, Germany). The VTIQ algorithm is employed to estimate the velocity of the shear waves induced by the ultrasound transducer, which correlates directly with tissue stiffness. The region of interest (ROI) for the measurements was specifically defined as the LN cortex. To ensure the accuracy of the measurements, a quality map was used to evaluate the reliability of each scan [32]. The quality map is an integral feature of the SWE system that visually represents the reliability of stiffness measurements across the ROI. The quality map helps identify areas where measurements may be less reliable due to factors such as probe movement or poor tissue contact. If the image quality was compromised due to excessive compression by the ultrasound probe or other factors, the measurement had to be repeated under more optimized conditions to avoid false results. For the SWE procedure, SLNs were fully embedded in ultrasound gel to ensure proper coupling and minimize any external artifacts. SWE was performed with minimum compression (on the gel) induced by the transducer. (Figure 1) Elasticity values were measured in meters per second (m/s), with a measurable range from 0 to 10 m/s. Elasticity values were measured nine times per LN.

### 2.3. Pathological Reference

Histopathological diagnosis was considered the reference standard. All pathological examinations of the SLNs were performed by dedicated breast pathologists following established national guidelines and standards to ensure consistency and reliability. The pathologists were blinded to the SWE findings, meaning they were unaware of the elastography results at the time of the histopathological analysis.

### 2.4. Statistical Analysis

This is an exploratory study. Statistical tests and resulting *p*-values can therefore only be interpreted descriptively. The characteristics of the study cohort were described by the measures of empirical distribution. Depending on the level of measurement, mean and standard deviation (SD) and absolute and relative frequencies were calculated. To compare the study cohort and the ultrasound morphology of the LNs with respect to their benign and malignant histopathology, unpaired t-tests were used. All statistical analyses were performed with R (version 04.1—© 2024, The R Foundation for Statistical Computing).

## 3. Results

### 3.1. Description of the Study Cohort and Shear Wave Elastography (SWE)

In total, 110 consecutive patients undergoing primary surgery were enrolled. Primary surgical therapy including an SLNB could be performed in all patients. Three patients had to be excluded because a correlation with the final histopathological finding was not possible; one patient showed reactive changes with silicone granuloma in the SLNB, and another patient showed lymphatic changes compatible with chronic lymphatic leukemia within the SLNB. Therefore, a total of 105 patients were enrolled for the statistical analysis. The mean age of the included patients was 63.69 ± 12.17 years. The breast cancer was localized in the right breast in 50 (47.62%) patients, and in the left breast in 55 (52.38%). Twelve patients (11.43%) had an in situ carcinoma in the core needle biopsy, of which three patients showed an invasive component in the final histology of the surgical specimen. The majority of patients had a cT1a-c (62.86%) or cT2 (20.95%) tumor stage, while five patients had a higher local tumor stage with cT3 (2.86%) or cT4 a, b (1.9%). With regard to the final pathological tumor stage, there was a slight shift towards higher stages, with sixty-seven patients with pT1a-c (63.81%), twenty-four patients with pT2 (22.86%), two patients with pT3 (2.86%), and three patients with pT4a, b (1.9%). A total of 71.43% of the breast cancer cases were an invasive ductal subtype, while 14.29% were invasive lobular. Six patients (5.72%) had other histological subtypes. A total of 85.42% of all invasive cancer cases were luminal-like, 6.25% were triple-positive, 1.04% were steroid-receptor negative, HER2-positive, and 7.29% were triple-negative. The cancer subtypes and the clinical and pathological tumor stages of the enrolled patients are summarized in Table 1.

In total, 168 SLNs were resected. The mean number of the excised SLNs per patient was 1.58.

Each LN was individually assessed with B-mode ultrasound and SWE. The mean size of the SLN was 9.56 ± 3.85 mm in the longest diameter and 4.84 ± 1.63 mm in the shortest diameter. SWE was performed within all 168 LNs. An example measurement is shown in Figure 2.

### 3.2. Pathology

Pathological examination of the SLNs was performed for all 168 LNs. LN metastases were detected in 17 LNs in 17 patients (16.19%). A total of 151 LNs in 88 patients were nonmetastatic (83.81%). The mean size of the metastatic LNs was 8.8 ± 3.51 mm in the longest diameter and 5.15 ± 1.91 mm in the shortest diameter, while benign LNs measured 9.64 ± 3.89 mm in the longest diameter and 4.8 ± 1.6 mm in the shortest diameter. Metastatic LNs were not significantly larger than nonmetastatic LNs (*p* = 0.442). The mean size of the metastasis was 5 ± 3.5 mm. The mean stiffness measured by SWE was 1.33 ± 0.23 m/s in nonmetastatic and 1.35 ± 0.29 m/s in metastatic SLNs (*p* = 0.724). (Figure 3) The maximum stiffness in the nonmetastatic and metastatic groups were 1.56 ± 0.38 m/s and 1.67 ± 0.45 m/s, respectively (*p* = 0.268). The minimum stiffness in both groups was 1.13 ± 0.17 m/s and 1.16 ± 0.24 m/s, respectively (*p* = 0.101). There was no significant difference between both groups regarding elasticity values.

## 4. Discussion

The confirmation of axillary LN metastasis is a critical factor in the prognosis of patients with early-stage breast cancer, significantly influencing treatment decisions and long-term outcomes. This study aimed to evaluate whether SWE, when performed in a standardized ex vivo setting, could reliably predict the presence of metastasis in SLNs. Previous research has demonstrated the potential of SWE for axillary LN staging, indicating that it can effectively distinguish between benign and malignant LNs as part of the pre-surgical diagnostic process. These studies have found that malignant LNs exhibit significantly higher shear wave velocities measured by SWE compared to benign LN. However, while in vivo SWE has shown promise, a few studies have explored whether ex vivo SWE on axillary LNs might be even more effective than its in vivo application for differentiating between benign and malignant LNs. The ex vivo setting allows a highly standardized condition, which helps to mitigate factors that can affect the accuracy and can introduce bias such as the individual application of pressure by the ultrasound probe, the depth of the LN within the axillary tissue, and anatomical complexities like adjacent blood vessels and bones. By enabling the real-time assessment of excised tissues during surgery, ex vivo SWE could potentially streamline workflows and improve patient care by minimizing delays in treatment.

Bae et al. conducted a study examining 228 axillary LNs in 55 breast cancer patients, including SLNs, non-SLNs, and other axillary LNs. Of the 55 patients, 10 had preoperatively known LN involvement and underwent ALND, while the other patients underwent an SLNB. In this cohort, 41 LNs were found to be metastatic, while 187 were tumor-free LNs. A significant correlation between the elasticity values measured by SWE and nodal metastasis was identified. Specifically, the mean stiffness in metastatic LNs was 45.4 kPA, whereas tumor-free LNs showed a mean stiffness of 17.7 kPA [31]. These values can be converted using the simplified formula for stiffness in kPa = 3 × (velocity in m/s)^2^ to 3.89 m/s and 2.43 m/s, respectively, demonstrating that metastatic LNs were significantly stiffer in this cohort [33,34].

Kilic et al. demonstrated in a cohort of 64 SLNs from 30 patients that cortical stiffness was significantly higher in malignant SLNs (n = 12, 18.8%) than in benign ones (n = 52, 81.2%). In this study, the mean stiffness in malignant SLNs was 25.5 kPA (converted to 2.92 m/s), whereas benign SLNs had a mean stiffness of 10.7 kPA (converted to 1.89 m/s) [30].

These two studies showed comparable results regarding significantly higher stiffness measured in metastatic LNs, although it must be noted that the examined cohorts are not directly comparable. Nonetheless, these findings are consistent with the current literature, which indicates that metastatic axillary LN seems to have higher elasticity values in vivo [24,25,26,35,36,37,38,39].

This aimed to evaluate a cleanly selected cohort focusing exclusively on SLNs through a direct one-to-one comparison between ex vivo elastography and the corresponding pathologic results under standardized conditions. Our cohort represents a low-risk group of patients with early-stage breast cancer, prospectively identified in a routine treatment setting in a specialized breast cancer unit. However, within this cohort, no significant difference in velocity was observed between metastatic and tumor-free SLNs. Additionally, the elasticity values measured in this study were notably lower than those reported in both ex vivo and in vivo studies found in the literature. In this cohort, the mean elasticity values were 1.33 ± 0.23 m/s in nonmetastatic and 1.35 ± 0.29 m/s in metastatic SLNs. In contrast, Bae et al. reported mean values of 3.89 m/s and 2.43 m/s while Kilic et al. reported 2.92 m/s and 1.89 m/s, respectively. In vivo measurements reported in the literature indicate mean elasticity values ranging from 2.41 to 4.32 m/s for malignant LNs and 1.64 to 2.37 m/s for benign LNs [24,25,26,35,36,37,38,39].

Variability in experimental settings, including probe pressure and SWE system configurations, may have contributed to these differences. Additionally, the patient cohort in this study may differ in clinical or pathological characteristics, which could influence SWE measurements. One important possible reason for the discrepancy in findings might be the size of the metastasis. In this study cohort, 17.7% of LNs were found to contain metastasis, with a mean metastasis size of 5 ± 3.5 mm. The mean size of the metastasis in the studies by Bae and Kilic discussed above is not explicitly mentioned. However, considering that in Bae et al.’s study, 10 out of 55 patients had preoperatively known LN involvement, the metastasis size may have been larger and possibly more easily detectable via SWE than in the current study. This raises the question of whether there is a technical threshold for the size of tissue that can be differentiated from the surrounding tissue with the current technical status of ultrasound elastography [40]. The placement of the ROI during SWE may also play a role in these discrepancies. If the ROI had not been accurately placed over the area of metastasis within the SLN, the elasticity values could have been underestimated. User error, particularly in identifying and targeting the exact site of the metastasis, may further contribute to the lack of differentiation observed in this cohort.

In this context, recent advancements in technology, particularly the integration of artificial intelligence (AI) in the interpretation of SWE measurements, have shown the potential to improve diagnostic accuracy by enabling more precise analysis, such as identifying stiffness patterns for better ROI placement, which may be challenging to detect manually [41,42,43,44].

Additionally, AI offers promising opportunities to enhance the understanding of SWE results when integrated into breast cancer pathology. AI-driven tools may provide a more precise and automated analysis of tumor characteristics, including heterogeneity and structural features, which may correlate with SWE measurements [45,46,47]. Incorporating such technological insights into future research could help bridge the gap between imaging findings and underlying tumor biology, improving the diagnostic utility of SWE in breast cancer.

There are several limitations to consider. The limited sample size in the LN metastases group restricts the statistical power and prohibits a generalization of the results. Another limitation of this study is the small tumor burden including micrometastases observed in the SLNs. Small tumor volumes may be characteristic of SLN metastases and could contribute to the lack of significant differences in SWE measurements between tumor-free and metastatic LNs. One important limitation is that tissue stiffness may be influenced by surgical resection, for example, due to the loss of perfusion and surrounding structural support. Ex vivo SWE reflects the mechanical properties of isolated tissue, which might differ slightly from in vivo conditions because of these environmental changes. While this limitation is inherent to ex vivo studies, it allows for controlled measurements without external confounding factors such as patient movement or variations in individual anatomy. Additionally, as is true for sonography per se, SWE underlies subjective observer bias, which cannot be ruled out as a limitation of the present study as well.

## 5. Conclusions

Under maximum standardized conditions, ex vivo SWE was unable to effectively discriminate between tumor-free and metastatic SLNs in this patient collective. One possible explanation for this outcome is that metastases present in SLN may predominantly be subclinical metastases and therefore cannot be adequately traced sonomorphologically. Given the current technical limitations of SWE, these subtle or early-stage metastatic changes might not result in sufficient alterations in tissue stiffness to be captured reliably by this imaging technique.

Further investigations are necessary to determine the full potential of SWE for non-invasive axillary staging, especially in low-risk patient cohorts where the detection of early metastases is critical for guiding treatment decisions. Larger studies with diverse patient populations could help identify specific cohorts where SWE is the most effective. These studies could explore improvements in SWE technology, a one-to-one comparison of velocities measured in vivo and ex vivo, integration with other imaging modalities, or its use in conjunction with biopsy methods to enhance diagnostic performance in breast cancer staging.

## Figures and Tables

**Figure 1 cancers-16-04270-f001:**
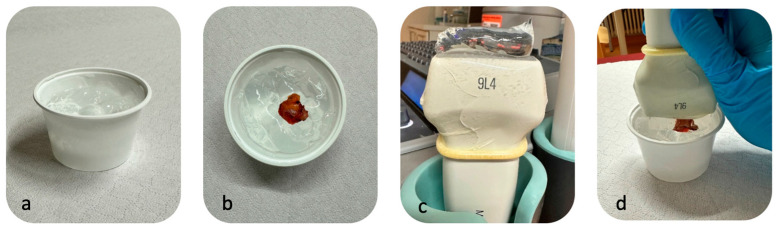
Ex vivo ultrasound scanning of a sentinel lymph node specimen. (**a**) Cup filled with ultrasound gel, avoiding air bubbles. (**b**) Sentinel lymph node specimen embedded in the ultrasound gel. (**c**) Ultrasound probe with additional gel to fully cover the sentinel lymph node specimen. (**d**) Performance of shear wave elastography with minimum compression.

**Figure 2 cancers-16-04270-f002:**
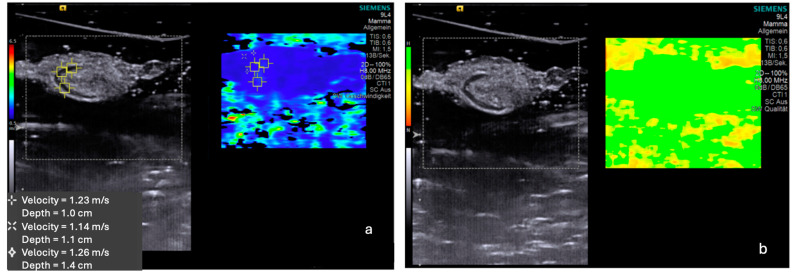
Example measurement of the shear wave elastography (**a**) and quality map (**b**) of an ex vivo sentinel lymph node. (**a**) Image of a single sentinel lymph node with three shear wave measurements. (**b**) Associated quality map of the same shear wave measurement. The area of the lymph node is completely green, indicating good quality of the shear wave measurement.

**Figure 3 cancers-16-04270-f003:**
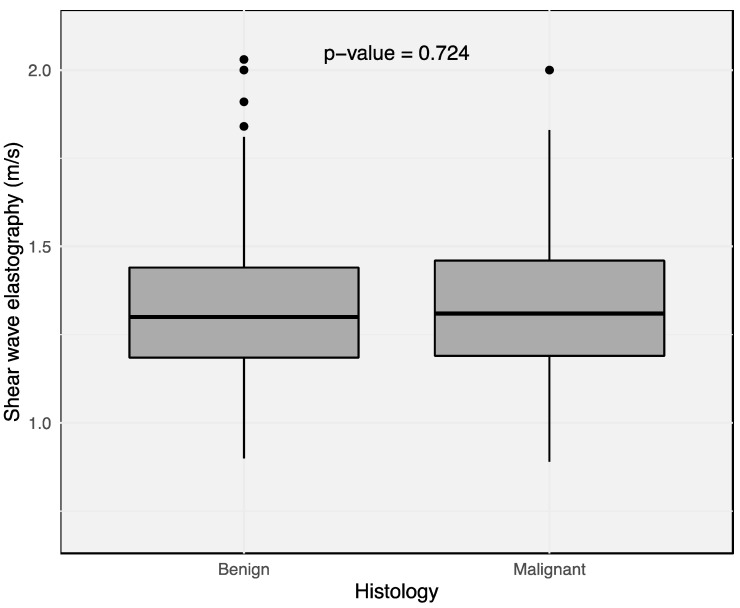
Velocities measured (m/s) by shear wave elastography in benign and malignant sentinel lymph node tissue. Mean stiffness in nonmetastatic lymph nodes was 1.33 ± 0,23 m/s and 1.35 ± 0.29 m/s in metastatic lymph nodes. There was no significant difference (*p* = 0.724).

**Table 1 cancers-16-04270-t001:** Baseline clinicopathologic characteristics of the study cohort.

Characteristics	Parameter	Absolute no.	%
Age	Mean (years ± SD)	63.69 ± 12.17	-
Laterality	Right	50	47.62
	Left	55	52.38
Tumor subtype	DCIS *	9	8.57
	IDC ^#^	75	71.43
	ILC **	15	14.29
	Other	6	5.71
Immunhistochemistry	Luminal-like	82	85.42
	HR ^##^ negative, HER2 ^†^ positive	1	1.04
	Triple-positive	6	6.25
	Triple-negative	7	7.29
Clinical tumor stadium	cTis	12	11.43
	cT1a–c	66	62.86
	cT2	22	20.95
	cT3	3	2.86
	cT4a,b	2	1.9
Pathological tumor stadium	pTis	9	8.57
	pT1a–c	67	63.81
	pT2	24	22.86
	pT3	2	1.90
	pT4a,b	3	2.86

* Ductal carcinoma in situ; ^#^ Invasive ductal carcinoma; ** Invasive lobular carcinoma; ^##^ Hormone receptor; ^†^ Human epidermal growth factor 2.

## Data Availability

The data presented in this study are available on reasonable request from the corresponding author due to ethical restrictions.

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
