# Peer review of "Evaluation of Ex Vivo Shear Wave Elastography of Axillary Sentinel Lymph Nodes in Patients with Early Breast Cancer"

_cancers, 2024, doi:10.3390/cancers16244270_

Round 1
Reviewer 1 Report
Comments and Suggestions for Authors
This study has pioneered a new approach where the ex vivo setting allows for highly standardized conditions, which helps to mitigate factors that may affect accuracy and potentially introduce bias, but there are still several issues that need further understanding.
- As a single-center prospective study, the number of positive samples is too small.
- The actual operation still needs to be performed in vivo, so it cannot be assumed that if there is no difference ex vivo, there will be no difference in vivo. It is suggested that, if possible, compare the differences between in vivo and ex vivo.
- The quantity and extent of tumor cells are still small, maybe just micrometastasis, so the differences are not significant.
- The statistical method needs to be corrected, and additional relevant factors and diagnostic accuracy need to be supplemented.
Author Response
Reviewer 1
Comments and Suggestions for Authors
This study has pioneered a new approach where the ex vivo setting allows for highly standardized conditions, which helps to mitigate factors that may affect accuracy and potentially introduce bias, but there are still several issues that need further understanding.
- As a single-center prospective study, the number of positive samples is too small.
Response: Thank you for this valuable comment. While we recognize that the sample size may limit the statistical power and generalizability of our findings, it reflects the real-world prevalence of sentinel lymph nodes metastases in early breast cancer patients with clinically negative nodal status. Additionally, the prospective design and rigorous methodology provide valuable insights into the application of shear wave elastography. We absolutely agree that future studies with larger cohorts are necessary to validate our findings and further assess the technique's clinical utility. This limitation has been addressed in the manuscript as an area for future research. (revised manuscript p. 8, line 432, line 454/455)
- The actual operation still needs to be performed in vivo, so it cannot be assumed that if there is no difference ex vivo, there will be no difference in vivo. It is suggested that, if possible, compare the differences between in vivo and ex vivo.
Response: Thank you for this valuable suggestion regarding the comparison between in vivo and ex vivo measurements. We agree that the results obtained ex vivo may not directly correlate with in vivo findings due to differences in tissue properties and physiological conditions. While our study focused on ex vivo analysis of sentinel lymph nodes, we acknowledge the importance of in vivo validation. Unfortunately, in vivo data was beyond the scope of this study. However, we believe that further research comparing both in vivo and ex vivo SWE measurements would be highly interesting and beneficial in assessing the clinical relevance of SWE for axillary staging. We will incorporate this suggestion as a direction for future studies in our revised manuscript. (revised manuscript p. 8, line 436 – 442, line 456/457)
- The quantity and extent of tumor cells are still small, maybe just micrometastasis, so the differences are not significant.
Response: Thank you for this valuable comment. We agree that the small quantity and limited extent of tumor cells, including micrometastases, may contribute to the lack of significant differences observed in SWE measurements. However, small tumor burden may be typical for metastases in sentinel lymph nodes. Future studies with a larger sample size and a focus on varying tumor burdens could help clarify the relationship between tumor extent and SWE values. Additionally, technical improvements in the SWE technique may enhance its sensitivity and ability to detect subtle differences associated with small tumor burdens. This aspect is now added to the discussion and limitation section of the revised manuscript. (revised manuscript, p.8 line 419 – 422, line 433- 436, line 454- 456)
- The statistical method needs to be corrected, and additional relevant factors and diagnostic accuracy need to be supplemented.
Response: Thank you for this valuable comment. We appreciate your observation regarding the statistical methods and the need to include additional relevant factors and diagnostic accuracy. In response, we have reviewed our statistical approach to ensure it is appropriate for the dataset and research objectives. However, due to the limited number of positive samples and the absence of a significant difference in SWE velocity between metastatic and tumor-free nodes, it is not possible to define an appropriate threshold for SWE velocity. Without a defined threshold, metrics such as diagnostic accuracy, sensitivity, specificity, positive predictive value, and negative predictive value cannot be reliably calculated. Future studies with larger cohorts and a broader range of tumor burdens will be necessary to understand the role of SWE in axillary staging in and ex vivo and to establish thresholds and evaluate the diagnostic accuracy of SWE in this context.
Reviewer 2 Report
Comments and Suggestions for Authors
The evaluation of ex vivo Shear-Wave Elastography of axillary sentinel lymph nodes in patients with early breast cancer, as proposed by the authors, is an interesting study. I have a few minor comments before publication:
1. For Figure 1, I recommend labeling each image and providing an explanation of what they represent.
2. In Figure 2, the labeling is unclear. It is difficult to distinguish which is "a" and which is "b." It would be better to clarify this further to ensure better understanding by the readers.
3. Please mention Figure 3 in the text and provide further explanation.
Author Response
Reviewer 2
Comments and Suggestions for Authors
The evaluation of ex vivo Shear-Wave Elastography of axillary sentinel lymph nodes in patients with early breast cancer, as proposed by the authors, is an interesting study. I have a few minor comments before publication:
- For Figure 1, I recommend labeling each image and providing an explanation of what they represent.
Response: Thank you for this recommendation. Each image has now been labelled and described individually.
- In Figure 2, the labeling is unclear. It is difficult to distinguish which is "a" and which is "b." It would be better to clarify this further to ensure better understanding by the readers.
Response: Thank you for this valuable comment. The images have now been labelled as “a” and “b”. Additionally, we have added detailed captions and explanations to ensure the figure is more informative and easier to understand for readers.
- Please mention Figure 3 in the text and provide further explanation.
Response: Thank you for this valuable comment. Figure 3 is mentioned in the manuscript line 320. Key data regarding mean stiffness values as well as the p-value are mentioned in the revised manuscript. These data are now added to the figure caption for better understanding.
Reviewer 3 Report
Comments and Suggestions for Authors
The manuscript presented a study to evaluate whether SWE can differentiate between tumor-free and metastatic-affected SLN ex vivo in a selected cohort. The work is well structured well and easy to follow.
The introduction started from a comparison of ALND and SLNB to introduce SWE and then SWE ex vivo and provided sufficient background. However, one major concern about this research is that the idea of running SWE ex vivo seems deviate from the idea of using SWE as a non-invasive assessment of tumors. In addition, if the tissue is already taken out and follow up histopathological examinations is conducted, what’s the value of running SWE ex vivo in such scenarios? Is the idea to apply SWE ex vivo without following examinations required in the future once the approach is established and standardized? There needs to be some discussion on the above two points.
Authors mentioned that having “greater control over variables such as probe pressure” as one of the advantages of SWE ex vivo in the introduction but it seems there is no further discussion related to the probe pressure or how it might affect the evaluation. Plus, if authors could provide some comparison between SWE in vivo against ex vivo, it could strengthen the argument.
In the methods section, it was mentioned that “measurement had to be repeated to avoid false results”. However, could stiffness be impacted during previous measurement and become unreliable in the repeated measurement? Furthermore, it would be helpful if authors could provide some discussion on whether stiffness can be changed upon removal of the tissue. One may argue if the SWE ex vivo truly reflects the stiffness changes due to tumor metastasis or environment change.
It could be helpful to include more details on the image process and data analysis. For example, how does the quality map work and how does it help? Is there any preprocess before the calculation of the metrics reported?
Are there some other different metrics that can be used to measure the Los in addition to the length in the longest and shorted diameter? For example, could there be a significant difference in the area? Could there be a significant difference in the distribution?
There could be some more in-depth discussion why the results from this study are contradictory to previous studies related to factors such as sample selection, experiment settings, patients’ cohort, etc. In addition, it would be better if authors could provide some potential directions for future investigation to address the limitation discussed. For example, how to further identify where the approach can be applied given there may or may not be significant differences among cohorts? Are there any approaches to reduce observer bias? Could there be a systematic way to automatically identify ROI?
Figure 1. Authors need to include more description for each panel to help readers understand what’s presented in the figure and better convey the message.
Figure 2 It would help readers better understand the figure if authors could provide some descriptions and explanations.
Author Response
Reviewer 3
Comments and Suggestions for Authors:
The manuscript presented a study to evaluate whether SWE can differentiate between tumor-free and metastatic-affected SLN ex vivo in a selected cohort. The work is well structured well and easy to follow.
The introduction started from a comparison of ALND and SLNB to introduce SWE and then SWE ex vivo and provided sufficient background. However, one major concern about this research is that the idea of running SWE ex vivo seems deviate from the idea of using SWE as a non-invasive assessment of tumors. In addition, if the tissue is already taken out and follow up histopathological examinations is conducted, what’s the value of running SWE ex vivo in such scenarios? Is the idea to apply SWE ex vivo without following examinations required in the future once the approach is established and standardized? There needs to be some discussion on the above two points.
Authors mentioned that having “greater control over variables such as probe pressure” as one of the advantages of SWE ex vivo in the introduction but it seems there is no further discussion related to the probe pressure or how it might affect the evaluation. Plus, if authors could provide some comparison between SWE in vivo against ex vivo, it could strengthen the argument.
Response: Thank you for raising these important points and for highlighting the need to address the rationale and future implications of applying SWE ex vivo. While SWE is primarily recognized as a non-invasive tool for in vivo assessment, our study serves as a proof-of-concept study to evaluate the technique’s feasibility and potential diagnostic value. Running SWE ex vivo allows for controlled conditions and eliminates the influence of external factors, such as patient movement and tissue perfusion, which can affect in vivo measurements. However, we acknowledge that we did not state clearly how probe pressure specifically might affect SWE evaluations. Excessive or inconsistent probe pressure can artificially alter tissue stiffness. By performing SWE ex vivo, we hypothesized that this variability is minimized, providing a more standardized and reproducible assessment. (Revised manuscript p. 3 line 235 – 237)
Regarding the comparison between in vivo and ex vivo SWE, while our study did not include in vivo measurements, we agree that such comparisons would strengthen the argument for ex vivo SWE. Unfortunately, in vivo data was beyond the scope of this study. We will incorporate this suggestion as a direction for future studies in our revised manuscript. (Revised manuscript p. 8 line 474/475)
Furthermore, we agree that the clinical utility of ex vivo SWE would ultimately depend on its ability to replace or complement histopathological analysis. Based on the current data, including the data collected in this study, this is not yet foreseeable. Future advancements may enable SWE to provide rapid, real-time assessments of excised tissues, potentially reducing reliance on time-intensive histopathological examinations in select cases. (revised manuscript p. 7 line 364 – 367)
We have expanded the discussion in the revised manuscript to address these points and clarify the potential implications of ex vivo SWE.
In the methods section, it was mentioned that “measurement had to be repeated to avoid false results”. However, could stiffness be impacted during previous measurement and become unreliable in the repeated measurement? Furthermore, it would be helpful if authors could provide some discussion on whether stiffness can be changed upon removal of the tissue. One may argue if the SWE ex vivo truly reflects the stiffness changes due to tumor metastasis or environment change.
Response: Thank you for this valuable comment. Regarding repeated measurements, we ensured that probe pressure and handling were consistent across measurements to minimize any potential alterations to tissue stiffness. To the best of our knowledge, there is currently no data available suggesting that repeated SWE measurements could alter tissue stiffness. However, we acknowledge that this is a potential limitation, particularly for soft tissues, and it warrants further investigation in future studies.
As for the impact of tissue removal, it is true that the stiffness of lymph nodes may be influenced by the loss of perfusion and surrounding structural support. Ex vivo SWE reflects the mechanical properties of isolated tissue, which might differ slightly from in vivo conditions due to these environmental changes. While this limitation is inherent to ex vivo studies, it allows for controlled measurements without external confounding factors such as patient movement or individual anatomy. We agree that a direct comparisons of in vivo and ex vivo measurements are essential for future studies. Such comparisons will provide valuable insights into the relationship between the two approaches and help determine whether SWE ex vivo can reliably reflect tumor-induced stiffness changes observed in vivo. This discussion has been included in the revised manuscript to emphasize this important area for further research. (revised manuscript p. 8 line 433 – 439, line 453/454)
It could be helpful to include more details on the image process and data analysis. For example, how does the quality map work and how does it help? Is there any preprocess before the calculation of the metrics reported?
Response: Thank you for this important suggestion to provide more details on the image processing and data analysis. In our study, we did not apply any preprocessing before the calculation of the reported metrics. The analysis was performed directly on the raw SWE data to preserve the integrity of the original measurements.
Regarding the quality map, it is an integral feature of the SWE system that visually represents the reliability of stiffness measurements across the region of interest. The quality map helps identify areas where measurements may be less reliable due to factors such as probe movement or poor tissue contact, allowing us to exclude these regions from the analysis. This ensures that only high-quality data is included in the calculation of metrics, thereby improving the accuracy and consistency of the results. We have added this explanation to the revised manuscript for clarity. (revised manuscript p. 3 line 251 – 255)
Are there some other different metrics that can be used to measure the Los in addition to the length in the longest and shorted diameter? For example, could there be a significant difference in the area? Could there be a significant difference in the distribution?
Response: Thank you for this insightful comment. While length in the longest and shortest diameters was used in our study to describe the lymph node, we agree that other metrics, such as area or distribution, could provide additional insights. Calculation of lymph node area or distribution of tissue properties within the lymph nodes could indeed be an important factor. However, this would require advanced imaging techniques and analysis methods, which are unfortunately beyond the scope of this study.
There could be some more in-depth discussion why the results from this study are contradictory to previous studies related to factors such as sample selection, experiment settings, patients’ cohort, etc. In addition, it would be better if authors could provide some potential directions for future investigation to address the limitation discussed. For example, how to further identify where the approach can be applied given there may or may not be significant differences among cohorts? Are there any approaches to reduce observer bias? Could there be a systematic way to automatically identify ROI?
Response: Thank you for this valuable comment. The discrepancies between our results and previous studies could stem from several factors. Differences in sample selection, such as differences in tumor burden, may have reduced the stiffness variations detectable by SWE. Variability in experimental settings, including probe pressure and SWE system configurations, may also have contributed. Additionally, the patient cohort in this study may differ in clinical or pathological characteristics, which could influence SWE measurements. (revised manuscript p.7, line 401 – 404, p.8 line 433- 436)
For future research, we propose several directions to address these limitations. First, larger studies with diverse patient populations could help identify specific cohorts where SWE is most effective. Second, implementing approaches to reduce observer bias, such as automated SWE image acquisition protocols and standardized guidelines for ROI selection, could improve reliability. AI-based systems to systematically and automatically identify ROIs may enhance consistency and reduce variability across studies. These advancements could help refine the clinical applicability of SWE for lymph node assessment. (revised manuscript p.8, line 454 – 457)
These points have been included in the revised discussion to provide a more comprehensive perspective on the study’s limitations and future directions.
Figure 1. Authors need to include more description for each panel to help readers understand what’s presented in the figure and better convey the message.
Response: Thank you for this recommendation. Each image has now been labelled and described individually.
Figure 2 It would help readers better understand the figure if authors could provide some descriptions and explanations.
Response: Thank you for this recommendation. In the revised manuscript, we have added detailed captions and explanations to ensure the figure is more informative and easier to understand for readers.
Reviewer 4 Report
Comments and Suggestions for Authors
The study provides an insightful exploration of the use of ex vivo Shear Wave Elastography (SWE) for assessing sentinel lymph nodes (SLNs) in early breast cancer patients. The methodology is well-explained, and the results are presented with clarity, making it a valuable contribution to the field. The authors have done a commendable job of outlining the limitations of their findings, especially in light of contrary results from previous studies.
However, I would recommend the authors consider enhancing the discussion by incorporating literature on breast cancer pathology, as it plays a pivotal role in understanding the biological underpinnings of SWE findings. Including recent advancements in pathology, particularly work published in 2024, could add depth and context to the paper. For instance, the authors might refer to studies such as:
- https://doi.org/10.3390/jcm9030749
Highlighting state-of-the-art research in breast cancer pathology can strengthen the study by providing a comprehensive overview of how SWE could complement existing pathological techniques. This would also help bridge the gap between imaging modalities and histopathological findings, emphasizing the clinical relevance of the study.
In summary, this is a well-conducted study, and with the suggested additions, it has the potential to make an even greater impact on the field of breast cancer research and axillary staging.
Author Response
Reviewer 4
Comments and Suggestions for Authors
The study provides an insightful exploration of the use of ex vivo Shear Wave Elastography (SWE) for assessing sentinel lymph nodes (SLNs) in early breast cancer patients. The methodology is well-explained, and the results are presented with clarity, making it a valuable contribution to the field. The authors have done a commendable job of outlining the limitations of their findings, especially in light of contrary results from previous studies.
However, I would recommend the authors consider enhancing the discussion by incorporating literature on breast cancer pathology, as it plays a pivotal role in understanding the biological underpinnings of SWE findings. Including recent advancements in pathology, particularly work published in 2024, could add depth and context to the paper. For instance, the authors might refer to studies such as:
- https://doi.org/10.3390/jcm9030749
Highlighting state-of-the-art research in breast cancer pathology can strengthen the study by providing a comprehensive overview of how SWE could complement existing pathological techniques. This would also help bridge the gap between imaging modalities and histopathological findings, emphasizing the clinical relevance of the study.
In summary, this is a well-conducted study, and with the suggested additions, it has the potential to make an even greater impact on the field of breast cancer research and axillary staging.
Response: Thank you for this suggestion. We agree that understanding the biological underpinnings of SWE findings is essential to contextualizing our results. In response, we expanded the discussion section and included recent advancements in breast cancer pathology including the above mentioned publication, to provide a deeper insight into the relationship between tumor biology and SWE measurements. This addition will strengthen the discussion and connect our findings to broader clinical and pathological frameworks. (revised manuscript Page 8 Line 441 – 447)
“Additionally, AI offers promising opportunities to enhance the understanding of SWE result when integrated in breast cancer pathology. AI-driven tools may provide more precise and automated analysis of tumor characteristics, including heterogeneity and structural features, which may correlate with SWE measurements. Incorporating such technological insights into future research could help bridge the gap between imaging findings and underlying tumor biology, improving the diagnostic utility of SWE in breast cancer. “
Round 2
Reviewer 3 Report
Comments and Suggestions for Authors
The authors' response letter provided sufficient clarification and valuable discussions and the manuscript looks good to me. However, it is noted that some line numbers cited in the response are not consistent with the revised manuscript. Please double check the version is up to date.